# Caffeine Doses of 3 mg/kg Increase Unilateral and Bilateral Vertical Jump Outcomes in Elite Traditional Jiu-Jitsu Athletes

**DOI:** 10.3390/nu13051705

**Published:** 2021-05-18

**Authors:** María Merino Fernández, Carlos Ruiz-Moreno, Verónica Giráldez-Costas, Cristina Gonzalez-Millán, Michelle Matos-Duarte, Jorge Gutiérrez-Hellín, Jaime González-García

**Affiliations:** 1Exercise and Sport Sciences, Health Sciences Faculty, Francisco de Vitoria University, UFV, Bulding E, Ctra, M-515 Pozuelo-Majadahonda Km 1800, 28223 Madrid, Spain; m.merino.prof@ufv.es (M.M.F.); michelle.matos@ufv.es (M.M.-D.); jghuniversidad@gmail.com (J.G.-H.); 2Education and Health Faculty, Camilo José Cela University, 28692 Madrid, Spain; cruizm@ucjc.edu (C.R.-M.); veronicagiraldezc@gmail.com (V.G.-C.); 24crgonzalez9@gmail.com (C.G.-M.)

**Keywords:** power, reliability, RFD, eccentric, concentric, force-time, jump biomechanics

## Abstract

Caffeine increases vertical jump, although its effects on kinetics and kinematics during different phases of bilateral and unilateral jumps remain unclear. The aim of this study was to identify the effects of 3 mg/kg on kinetic, kinematic and temporal variables in the concentric and eccentric phases of bilateral and unilateral countermovement jumps. A total of 16 Spanish national team traditional Jiu-Jitsu athletes took part in two experimental trials (3 mg/kg caffeine or placebo) in a randomized, double-blind crossover study. Sixty minutes after ingestion, bilateral and unilateral jumps were performed on a force platform. Compared to the placebo, caffeine increased bilateral jump height (*p* = 0.008; Δ% = 4.40), flight time (*p* = 0.008; Δ% = 2.20), flight time:contraction time (*p* = 0.029; Δ% = 8.90), concentric impulse (*p* = 0.018; Δ% = 1.80), peak power (*p* = 0.049; Δ% = 2.50), RSI-modified (*p* = 0.011; Δ% = 11.50) and eccentric mean braking force (*p* = 0.045; Δ% = 4.00). Additionally, caffeine increased unilateral RSI-mod in both legs (Left: *p* = 0.034; Δ% = 7.65; Right: *p* = 0.004; Δ% = 11.83), left leg flight time (*p* = 0.044; Δ% = 1.91), left leg jump height (*p* = 0.039; Δ% = 3.75) and right leg FT:CT (*p* = 0.040; Δ% = 9.72). Caffeine in a dose of 3 mg/kg BM in elite Jiu-Jitsu athletes is a recommended ergogenic aid as it increased performance of bilateral and unilateral vertical jumps. These increases were also accompanied by modified jump execution during the different phases of the countermovement prior to take-off.

## 1. Introduction

Caffeine (1,3,7-trimethylxanthine) is a psychoactive substance found in many different sports supplements due to its associated ergogenic effect when ingested 45–90 min before sports practice [1]. Given its performance enhancing effects, this substance is recurrently used by different sports athletes and has been found in 74% of the urine samples analyzed in doping controls [2]. It is accepted that the main mechanism associated with sports performance is produced in the central nervous system (CNS) as an antagonist of the adenosine receptors due to its similar molecular form [1]. However, the mechanism related to caffeine in anaerobic power movements seems inconclusive. It is suggested that its effect in a peak power test could be through tetanic tension [3], however, high and toxic concentrations of caffeine in humans are necessary to produce the aforementioned neuromuscular process [3,4]. One of the increased power events associated with caffeine consumption is countermovement jump (CMJ) height [5,6,7,8,9], which is a useful and sensitive tool to monitor lower limb neuromuscular status [10]. However, only analyzing the main outcomes of the CMJ prevents us from understanding the neuromuscular function along the force-time curve and the different phases of the jump [11]. To give an example of this in-depth analysis, a longitudinal examination identified that training not only influenced the performance of the traditional variables (peak power, peak velocity and jump height), it also modified the shape of the power-, force-, velocity- and displacement-time curves during the different phases of the CMJ [12]. In addition to bilateral CMJ force-time curve or phase analysis, unilateral CMJ assessment is receiving more attention due to the claim that the unilateral jump presents more similarities (one-leg nature) with different sports actions such us running strides and changes of direction [13]. Therefore, a unilateral assessment may offer better training information in order to reproduce competition-specific movement patterns. In this respect, the single-leg vertical jump has been significantly related to 10 m acceleration performance, change of direction times, the single-leg lateral jump and single-leg horizontal jump [14].

Previous research has identified that moderate doses (3–5 mg/kg) of caffeine increase power output [8] and CMJ height by 3.76–4.3% in different athletes [5,15]. Bloms et al. [5] have reported that caffeine increases peak force and average rate of force development (RFD). However, in these studies only concentric phase metrics were evaluated, which could be a limited assessment of the whole jump since the stretch-shortening cycle has not been analyzed. Just one study presents insights into the effects of caffeine in all phases of a CMJ [6]. The authors observed that caffeine increases concentric phase kinetics and kinematics, however, caffeine did not modify any temporal or performance metrics during the eccentric phase, as similarly observed by Lago-Rodríguez et al., [9]. Despite the observed performance enhancing effects of a low dose of caffeine on bilateral CMJ, the hypothetic potential of caffeine to improve single-leg jump performance and the absence of side effects in comparison to higher doses (i.e., 6 and 9 mg/kg) [16], to the authors’ knowledge, no research has established the possible enhancing effects of 3 mg/kg of caffeine on bilateral and unilateral countermovement jump kinetics and kinematics in the different phases of the jump. Therefore, the aim of this research was twofold: (i) to identify the effects of 3 mg/kg on the performance and temporal outcomes of bilateral CMJ in all its phases and (ii) to establish the possible ergogenic effects of this dose on performance and temporal metrics of the unilateral CMJ. We hypothesized that 3 mg/kg of caffeine would increase the concentric metrics of both bilateral and unilateral CMJ kinetics and kinematics as this dose appears to increase high velocity-low resistance actions [16].

## 2. Materials and Methods

A double-blind, placebo-controlled, crossover design was used in this investigation to determine the possible ergogenic effects of 3 mg/kg of caffeine on performance, kinetics and kinematics of the bilateral and unilateral CMJ. The collected dependent variables included typical CMJ variables, alternative concentric and alternative eccentric metrics, as classified previously [17].

Sixteen (*n* = 16) elite traditional Jiu-Jitsu athletes were enrolled in this investigation. The participants were eight men (height = 176.87 ± 6.31 cm; height = 72.59 ± 10.11 kg; age = 21.50 ± 4.75 years; training experience = 11.88 ± 3.94 years; weekly training = 11.63 ± 1.85 h/week), and eight women (height = 165.63 ± 6.39 cm; weight = 64.86 ± 6.33 kg; age = 20.63 ± 3.20 years; training experience = 15.38 ± 2.92 years; weekly training = 11.75 ± 2.19 h/week). Due to a force platform software error analysis, just the results of 14 participants were included in the analysis for the unilateral jump. All the participants who fulfilled the following inclusion criteria: (a) aged between 18 and 35 years old; (b) caffeine naïve or low habitual caffeine consumers (<0.99 mg/kg/BM/day), were selected for participation. Subjects were excluded if they reported (a) medication usage within the previous month; (b) a previous history of cardiopulmonary diseases; or (c) use of oral contraceptive pills, as they may interfere with caffeine pharmacokinetics. Participants were informed about the experimental procedures and the possible risks and benefits associated with taking part in the investigation. Additionally, they signed the written informed consent form to participate in this research. The study and informed consent procedures were approved by the University Ethics Committee under the latest version of the Declaration of Helsinki.

### 2.1. Pre-Experimental Procedures

On two different days, participants carried out two familiarization sessions to establish inter-day reliability of the dependent variables. Both familiarization sessions were spaced 72 h apart, with 3 bilateral CMJ and 6 unilateral CMJ (3 with the left leg and 3 with the right leg) performed. The mean value of each three attempts was entered into the statistical analysis as averaged jumps were more sensitive than the highest jump in detecting fatigue or adaptations [10]. Participants rested three minutes between each attempt. All participants carried out the jumps at the same time of day. The same standardized warm-up was performed before each session which included 5 min of cycling with a rate of perceived exertion (RPE) of 5/10, 3 min of hip, knee and ankle mobility and 10 bodyweight squats [18]. Participants followed the instructions to jump as high as possible and received the same verbal encouragement during the trials.

### 2.2. Experimental Procedures

Each athlete carried out two identical experimental trials, separated by a week to allow complete recovery and substance wash-out. Each participant performed 3 CMJ with both legs, 3 CMJ with the right leg and 3 CMJ with the left leg on a force platform, and 3 min of rest were programmed between each attempt. Participants ingested an unidentifiable capsule with either caffeine anhydrous (3 mg/kg·bm, Bulk Powders 100% purity, Colchester, UK) or an inert substance (Placebo; cellulose, Guinama, Valencia, Spain) 60 min before data collection. This time frame is adequate for maximizing blood caffeine concentration [19]. This dose was selected due to the benefits of increasing lower limb mechanical performance without increasing the possible side effects associated with caffeine consumption [15]. The capsules were prepared by an external researcher to ensure the blinding protocol. An alphanumeric code was assigned to each trial to blind the trial condition to participants and researchers. This coding was not revealed until the data had been analyzed.

All jumps were performed on a ForceDecks FD4000 Dual Force Platform (ForceDecks, London, United Kingdom), with a sampling rate of 1000 Hz. The collected data from each jump was entered into the ForceDecks software (ForceDecks, London, United Kingdom) to analyze each jump and generate all of the dependent variables. The reliability of the dependent variables is presented in Table 1. The onset of the movement was determined when a drop of 20 N from measured weight was produced. Vertical jump was divided into eight key phases: weighing phase, eccentric phase, braking phase, deceleration phase, concentric phase, flight phase and landing phase. To be accurate in the weighing phase, at least one second of measurement was used [11]. The eccentric phase was determined as the jump phase with negative center of mass (COM) velocity. Additionally, the braking phase was the subphase within the eccentric phase, which lasted from the instant where minimum force was detected until the end of the eccentric phase. The deceleration phase started at the moment that negative peak velocity was achieved and finished at the end of the eccentric phase (COM minimum displacement). The concentric phase started at the moment when velocity became positive and ended at take-off. The flight phase lasted from take-off to the landing and the landing phase lasted from the point where the force rose above 30 N and returned to bodyweight (Figure 1). The collected dependent variables included in this research were established based on the classification by Heishman et al. [17].

### 2.3. Statistical Analysis

Statistical significance tests were carried out using IBM SPSS Statistics for Macintosh, Version 26.0 (IBM Corp., Armonk, NY, USA) and Microsoft Excel (2013; Microsoft Corporation, Albuquerque, NM, USA). The sample size was estimated using free software (G*Power v3.1). Sample size estimation revealed that 12 participants were sufficient for a one-tailed paired sample *t*-test with an effect size of 1.2. This effect size was calculated with the mean and SD of placebo and caffeine CMJ height from the previous literature and a Pearson correlation of 0.97 (data obtained during familiarization) assuming 5% type I and type II errors. A paired sample *t*-test was performed to identify the effects of 3 mg/kg of caffeine. If any dependent variable did not achieve the Shapiro-Wilk normality assumption, the Wilcoxon test was used. Between pairs, at 95% CI, Cohen’s *d* effect size (ES) was calculated using an excel spreadsheet. Intraclass Correlation Coefficient (ICC), Standard Error of the Measurement (SEM) and minimum detectable change (MDC) at 95% confidence interval (95% CI) [20] were calculated. ICC values were analyzed based on the following criteria: poor reliability, <0.5; moderate reliability, 0.5–0.75; good reliability, 0.75–0.90; and excellent reliability, >0.90 [21]. Cohen’s *d* estimated magnitudes were calculated and classified in all between-groups comparisons [22]. Results are expressed as mean ± standard deviation (SD). The significance level was set at *p* < 0.05.

## 3. Results

Most of the variables for bilateral and unilateral jumps presented more than acceptable reliability (Table 1).

### 3.1. Bilateral CMJ

The effects of caffeine on all of the bilateral CMJ metrics are displayed in Table 2. Caffeine consumption significantly increased jump height using the flight time method (*p* = 0.008; mean diff = 1.23 cm [0.27 to 2.18]; Δ% = 4.40 [0.90 to 8.00]). Flight time (*p* = 0.008; mean diff = 20.25 ms [2.30 to18.20]; Δ%= 2.20 [0.40 to 4.00]), flight time:contraction time (FT:CT) (*p* = 0.029; mean diff = 0.05 [0.00 to 0.11]; Δ% = 8.90 [−1.40 to 20.20]), concentric impulse (*p* = 0.018; mean diff = 2.83 Ns [0.22 to 5.45]; Δ% = 1.80 [0.20 to 3.50]), absolute peak power (*p* = 0.049; mean diff = 75.47 W [−15.75 to 166.69]; Δ% = 2.50 [−0.50 to 5.50]) and Reactive Strength Index-modified (RSI-Modified) (*p* = 0.011; mean diff = 0.04 ms [0.01 to 0.08]; Δ% = 11.50 [0.90 to 23.30]) were also higher in the caffeine condition. Additionally, jump height calculated using the impulse-momentum method tended to be higher after caffeine consumption (*p* = 0.059; mean diff = 0.8 cm [−0.23 to 1.83]; Δ% = 2.70 [−0.90 to 6.40]). The mean eccentric force produced in the braking phase was higher in the caffeine condition (*p* = 0.045; mean diff = 30.58 [−5.23 to 66.40]; Δ% = 4.00 [−0.30 to 8.40]). Despite no significant results (*p* > 0.05), most of the kinetic and kinematic metrics of the concentric and eccentric phases presented small to moderate ES in the caffeine condition.

### 3.2. Unilateral CMJ

The effects of caffeine on all of the unilateral CMJ metrics are displayed in Table 3. As depicted in Figure 2 (Panels D–F), the RSI-modified were higher in both legs after caffeine consumption (Left leg: *p* = 0.034; mean diff = 0.012 ms [−0.00 to 0.03]; Δ% = 7.65 [−0.30 to 16.23]; right leg: *p* = 0.004; mean diff = 0.017 ms [−0.00 to 0.04]; Δ% = 11.83 [0.07 to 24.97]. Flight time (*p* = 0.044; mean diff = 7.5 ms [−1.26 to 16.26]; Δ% = 1.91 [−0.72 to 4.60]) and jump height (flight time method) (*p* = 0.039; mean diff = 0.71 cm [−0.09 to 1.52]; Δ% = 3.75 [−1.61 to 9.39]) were higher for the left leg in the caffeine condition. Moreover, FT:CT was also higher in the caffeine condition only in the right leg CMJ (*p* = 0.040; mean diff = 0.02 [−0.01 to 0.05]; Δ% = 9.72 [−0.32 to 20.77]). No differences between placebo and caffeine were observed in any other kinetic or kinematic metrics (*p* > 0.05).

## 4. Discussion

The aims of the present study were to identify the effects of 3 mg/kg of caffeine on jump performance, kinetics and kinematics of the bilateral and unilateral CMJ in all their phases in elite Jiu-Jitsu athletes. As was argued, low to high doses of caffeine enhance CMJ height and lower limb performance [5,6,8,15,23,24]. In fact, our results indicated that eccentric mean braking force, peak power, concentric impulse and flight time in the bilateral CMJ were higher after caffeine intake which translated into greater jump height. These increases were accompanied by different jump execution as explained by the differences between placebo and caffeine conditions in the RSI-mod and FT:CT. The results of the unilateral CMJ were similar, showing increases in RSI-mod, flight time, jump height and FT:CT. To the authors’ knowledge, this is the first research to highlight the SEM and MDC of the unilateral CMJ-related variables in a top-level combat sport cohort. 

Our results showed increases of 2.7% (0.8 cm) to 4.4% (1.23 cm) in jump height, depending on the calculation method, following the consumption of 3 mg/kg of caffeine. These results agree with previous literature that used low to moderate doses of caffeine as an ergogenic aid [5,8,9,15]. In that research, the acute consumption of caffeine in volleyball, handball and collegiate athletes increased from 3.4 to 4.7%. Following a deeper analysis, our data revealed that concentric mean force increased by 2.50% [−1.70 to 6.80], concentric mean power by 3.40% [−2.20 to 9.10] and peak power by 2.50% [−0.50 to 5.50] (Figure 2, Panel A). These outcomes from the concentric phase of the jump were lower in comparison to those observed in previous research where concentric force increased by 6.5 ± 6.4%, peak force by 4.9 ± 9.6% [5] and peak power by 16.2 ± 8.3% [6]. These higher effects on concentric performance may be explained by the different dose of caffeine administered (3 vs. 5 mg/kg), or the different sports backgrounds; although doses ranging from 2–6 mg/kg of caffeine have shown ergogenic effects in the vertical jump test in recreationally active males [25]. It is suggested that the observed ergogenic effects are mediated by the CNS, as caffeine affects central and peripheral pathways which would induce stimulation of motor neurons, resulting in an increased release of Ca^2+^ from the sarcoplasmic reticulum [4]. This could generate a better electrochemical gradient which would induce efficient contractile activity of the effector muscle [26]. 

To our knowledge, only one study has examined the effects of caffeine on the eccentric phase of CMJ and it showed no beneficial effects on the eccentric peak force or the eccentric RFD [6]. Equally, our results failed to reveal any differences in mean and peak eccentric force. However, mean force and RFD during the braking phase were higher in the caffeine condition (Figure 2, Panel C). Previous isokinetic based research identified that the increases in torque production, rate of torque development and normalized muscle activities were not dependent on the type of contraction [27]. We therefore believe that it is necessary to further research the effects of caffeine on the kinetics of the eccentric phase of the jump. 

In parallel, jumps in the caffeine condition presented a different execution than with the placebo. Specifically, the duration of the braking phase was shortened by 11.4% after caffeine intake and was accompanied by an increase of 4% in the eccentric force, which was translated into a higher braking RFD (Figure 2, Panel C). The increased force production in the eccentric phase may influence the force production in the concentric phase by using a better strategy performed during the eccentric phase of the jump. A greater unloading phase, may have allowed higher negative momentum which was possibly effectively converted into force, leading to higher concentric outcomes [28]. Our data also revealed that 12 of the 14 participants reduced the total contraction time (not only a reduction in the braking phase) in the caffeine condition without a decrease in concentric force production and jump height (Figure 2 Panels A and B). Consequently, a higher RSI-mod was observed in the caffeine condition (Δ% = 11.50 [0.90 to 23.30]). Previous research did not observe differences in the duration of the jump after caffeine consumption [5,6]. However, this shortened jump execution seems to be effective for our participants as concentric net impulse and jump height increased significantly despite the lower contraction times (Table 2 and Figure 2 panel A). The shortened jump execution may be produced by possible decreased tiredness, mood improvement and higher energetic arousal, as a result of the effects of caffeine on the central neural system [29]. 

This is the first study that has aimed to identify the effects of 3 mg/kg of caffeine on unilateral CMJ performance during the different phases of the jump. Contrary to the bilateral jump, no improvements were observed in the kinetic or kinematic parameters of the eccentric phase of the unilateral jumps following caffeine intake. In fact, lower mean eccentric force and shortened deceleration and braking phases were observed in the caffeine condition in comparison to the placebo (Figure 2, Panel F). Although the kinetics of the eccentric phase partly explains the variance in vertical jump performance [30], the jump height after caffeine consumption increased by 1.30 cm [−0.95–3.55] for the left leg and by 0.80 cm [−1.03–2.63] for the right (Figure 2, Panel D). It is suggested that the increased vertical displacement could be produced by the effects of caffeine on the concentric phase as the outcomes in this phase were moderately to highly correlated with vertical jump displacement [31]. Despite not being statistically significant, caffeine intake produced small increases (ES > 0.2) in concentric mean and peak force which led to higher acceleration and velocity which ultimately resulted in a higher jump [32] (Figure 2, Panels D and E). 

Just RSI-mod was statistically higher after caffeine consumption for both unilateral jumps, increasing by 7.65% for the left leg and 11.83% for the right CMJ (Figure 2, Panel D). However, increases in the RSI-mod occurred in different ways for each leg. For the left leg, the increases in the RSI-mod were mainly explained by greater jump height (+1.3 cm; ES = 0.54) accompanied by a slight reduction in the contraction time (−23.84 ms; ES = −0.21), while the increases in the RSI-mod for the right leg were produced by lower contraction time (−56.92 ms; ES = −0.40) and a slight increase in jump height (+0.8 cm; ES = 0.25). However, increases in RSI-mod outcomes may have been produced not only by caffeine intake as the unilateral jump has demonstrated a variable ground reaction force distribution, which in turn, may increase the variability of the time-related variables [14]. 

Finally, this research presented different limitations that should be considered when interpreting the results: (i) We cannot categorically conclude that the observed changes were only produced by the caffeine intake as the calculated minimum detectable change is greater than any of the observed mean differences. (ii) The relative and absolute reliability observed for the bilateral jump in this cohort was lower than previously observed in NCAA Division 1 intercollegiate basketball players [17], suggesting that the CMJ test is not as sensitive to changes in cohorts with low experience in vertical jumping. (iii) Countermovement depth was self-selected by the participants which may increase intra-individual variability due to differences in jump execution. (iv) The study was only carried out with top-level traditional Jiu-Jitsu athletes and results may not transfer to different sports cohorts, lower-level athletes or the general population.

Previous studies have demonstrated the ergogenic capacity of caffeine in bilateral vertical jump performance in different populations [5,8,15] but not in unilateral ones. Additionally, its effects on the different jump phases remain unclear. Based on our results, caffeine is an effective ergogenic aid if the aim is to increase vertical jump performance since it enhanced bilateral and unilateral jump height following a shorter jump execution. This ability to produce greater jump heights in shorter contraction times has a direct implication for those sports that involve performing this task in a shorter time than the opponent. The present research demonstrates that caffeine increases force production in the eccentric and concentric phase of the bilateral jump which translates into higher jumps and increases unilateral jump height with shorter jump execution.

## 5. Conclusions

The intake of 3 mg/kg of body weight of caffeine in elite Jiu-Jitsu athletes is an ergogenic aid that increases the performance of both bilateral and unilateral vertical jumps, and also modifies jump execution along with the different phases of the countermovement.

## Figures and Tables

**Figure 1 nutrients-13-01705-f001:**
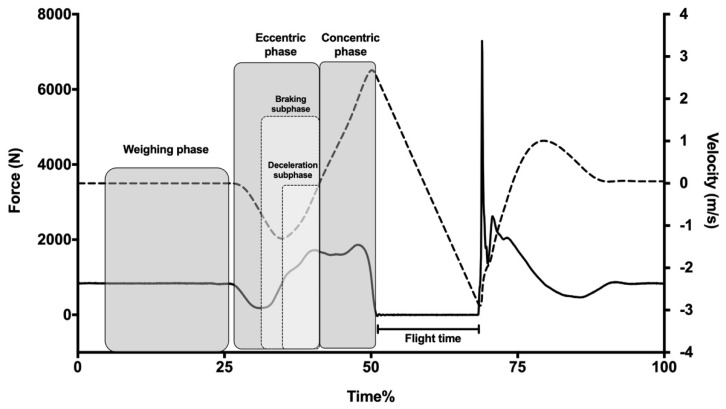
Force- and velocity-time curves of a representative participant.

**Figure 2 nutrients-13-01705-f002:**
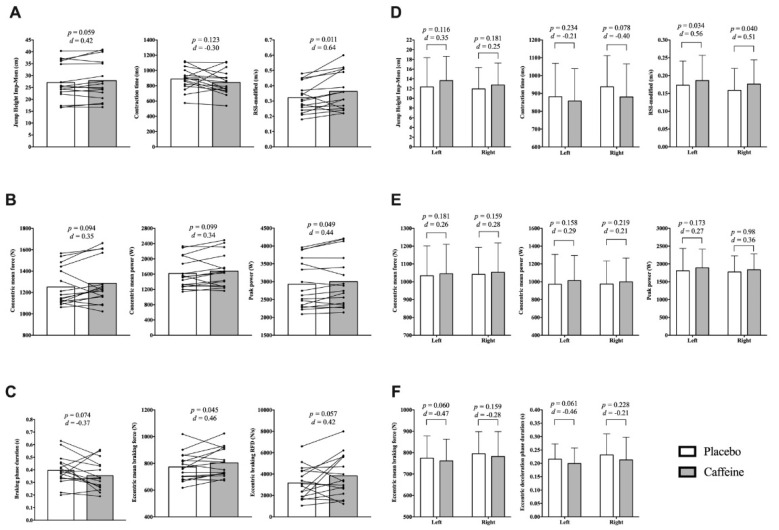
Mean, SD, ES and individual responses (**A**–**C**) between placebo and caffeine conditions in bilateral jump and mean, SD and ES between placebo and caffeine conditions in unilateral jumps in the different phases (**D**–**F**).

**Table 1 nutrients-13-01705-t001:** Reliability of the bilateral and unilateral CMJ variables.

	Bilateral			Left			Right		
	ICC	SEM	MDC CI 95%	ICC	SEM	MDC CI 95%	ICC	SEM	MDC CI 95%
Conc MF/BM [N/kg]	0.69	0.5	1.41	0.92	1.80	5.01	0.89	1.5	4.19
Conc MF [N]	0.98	35.70	98.73	0.99	19.58	54.27	0.97	30.96	85.83
Conc PF/BM [N/kg]	0.81	0.50	1.49	0.90	0.72	1.99	0.83	0.81	2.24
Conc PF [N]	0.99	34.90	96.61	0.96	49.32	136.70	0.93	61.51	170.48
Conc M Power/BM [W/kg]	0.88	0.90	2.58	0.93	0.94	2.61	0.62	1.81	5.01
Conc M Power [W]	0.98	57.20	158.68	0.97	54.33	150.59	0.77	126.83	351.57
FT [ms]	0.92	13.80	38.22	0.97	11.94	33.09	0.91	18.75	51.99
CT [ms]	0.76	62.29	172.65	0.85	74.67	206.98	0.82	105.49	292.40
FT:CT	0.76	0.03	0.08	0.82	0.04	0.12	0.56	0.07	0.19
JH (Flight Time) [cm]	0.96	1.46	4.04	0.97	1.04	2.88	0.89	1.71	4.73
JH (Imp-Mom) [cm]	0.97	1.48	4.08	0.93	1.82	5.06	0.56	3.09	8.56
P Power/BM [W/kg]	0.98	0.94	2.61	0.92	2.34	6.48	0.64	3.20	8.87
P Power [W]	0.99	67.93	188.31	0.96	136.87	379.37	0.78	222.73	617.39
Conc Dur [ms]	0.84	20.35	55.53	0.85	23.24	64.41	0.82	25.43	70.49
Conc Impulse [Ns]	0.99	4.35	12.07	0.97	6.03	16.72	0.78	12.92	35.82
Conc P Vel [m/s]	0.96	0.07	0.18	0.91	0.11	0.30	0.54	0.18	0.51
Conc RFD/BM [N/s/kg]	0.92	6.05	16.77	0.83	5.15	14.27	0.53	6.29	17.45
Conc RFD [N/s]	0.90	415.1	1150.5	0.88	327.0	906.3	0.65	434.5	1204.3
Conc RPD/BM [W/s/kg]	0.66	19.57	54.24	0.95	19.91	55.19	0.84	15.39	42.67
Conc RPD [W/s]	0.78	1395.9	3869.3	0.96	758.3	2102.0	0.86	1167.0	3234.6
F at P Power [N]	0.97	51.32	142.23	0.97	35.44	98.25	0.97	37.22	103.16
RSI-modified [m/s]	0.76	0.02	0.01	0.90	0.02	0.06	0.68	0.04	0.10
Vel at P Power [m/s]	0.94	0.08	0.22	0.91	0.10	0.27	0.50	0.17	0.46
Braking Phase Dur [s]	0.31	0.06	0.16	0.79	0.06	0.16	0.79	0.07	0.20
Ecc Braking RFD/BM [N/s/kg]	0.47	11.29	31.29	0.75	9.48	26.28	0.62	14.35	39.78
Ecc Braking RFD [N/s]	0.69	728.8	2020.2	0.76	660.8	1831.6	0.61	1062.5	2945.0
Ecc Dec Phase Dur [s]	0.44	0.04	0.12	0.75	0.04	0.10	0.80	0.06	0.16
Ecc Dec RFD/BM [N/s/kg]	0.48	25.02	69.36	0.90	7.53	20.88	0.74	15.07	41.77
Ecc Dec RFD [N/s]	0.56	1914.3	5306.1	0.91	509.8	1413.1	0.73	1087.0	3013.0
Ecc M Braking F [N]	0.95	35.87	99.42	0.94	28.70	79.56	0.86	49.12	136.16
Ecc M Dec F [N]	0.87	103.79	287.70	0.88	62.79	174.03	0.87	74.03	205.20
Ecc MF [N]	1.00	2.01	5.58	1.00	6.45	17.87	0.98	13.00	36.03
Ecc M Power/BM [W/kg]	0.48	0.75	2.08	0.70	0.47	1.31	0.52	0.68	1.89
Ecc M Power [W]	0.85	50.31	138.44	0.85	33.60	93.16	0.63	47.60	132.05
Ecc PF/BM [N/kg]	0.53	2.57	7.11	0.90	0.73	2.03	0.78	1.20	3.34
Ecc PF [N]	0.81	204.43	566.66	0.95	52.08	144.37	0.87	94.33	261.48
F at Zero Vel [N]	0.81	205.11	568.54	0.96	50.53	140.07	0.87	93.21	258.37

BM = Body mass; Conc MF = concentric mean force; Conc PF = concentric peak force; Conc M power = concentric mean power; FT = Flight time; CT = Contraction time; FT:CT = ratio between flight time and contraction time; JH = Jump height; P power = concentric peak power; Conc Dur = duration of the concentric phase; Conc P Vel = concentric peak velocity; Conc RFD = concentric rate of force development; Conc RPD = concentric rate of power development; F at P Power = Force at peak power; RSI-mod = Reactive strength index-modified; Vel at P Power = Velocity at peak power; Braking phase dur = braking phase duration; Ecc Braking RFD = rate of force development during braking phase; Ecc Dec Phase Dur = Duration of the eccentric deceleration phase; Ecc Dec RFD = rate of force development during deceleration phase; Ecc M Braking F = mean force during braking phase; Ecc M Dec F = mean force during deceleration phase; Ecc MF = Eccentric mean force; Ecc M Power = Eccentric mean power; Ecc PF = Eccentric peak force; F at Zero Vel = Force at zero velocity.

**Table 2 nutrients-13-01705-t002:** Kinetic and kinematic differences between Caffeine and Placebo conditions in the bilateral CMJ.

	Placebo		Caffeine				CI 95%	
	Mean	SD	Mean	SD	*p*	ES	Lower Limit	Upper Limit
Conc MF/BM [N/kg]	18.28	1.87	18.60	2.12	0.185	0.23	−0.27	0.72
Conc MF [N]	1251.44	176.15	1286.00	207.14	0.094	0.35	−0.17	0.85
Conc PF/BM [N/kg]	22.47	2.99	22.74	2.97	0.337	0.11	−0.39	0.60
Conc PF [N]	1551.09	279.07	1577.66	268.33	0.291	0.14	−0.35	0.63
Conc M Power/BM [W/kg]	23.39	4.85	24.18	5.46	0.131	0.29	−0.21	0.79
Conc M Power [W]	1617.31	395.33	1678.03	436.53	0.099	0.34	−0.17	0.84
FT [ms]	471.34	62.33	481.59	64.08	**0.008**	0.69	0.13	1.23
CT [ms]	888.47	145.83	843.78	157.71	0.123	−0.30	−0.80	0.20
FT:CT	0.55	0.12	0.60	0.15	**0.029**	0.52	−0.01	1.03
JH (Flight Time) [cm]	27.68	7.20	28.91	7.72	**0.008**	0.68	0.13	1.22
JH (Imp-Mom) [cm]	27.08	7.69	27.88	8.24	0.059	0.42	−0.10	0.92
P Power/BM [W/kg]	42.44	8.89	43.20	9.15	0.162	0.25	−0.25	0.75
P Power [W]	2927.97	676.85	3003.44	719.17	**0.049**	0.44	−0.08	0.95
Conc Dur [ms]	280.56	41.71	272.84	42.46	0.255	−0.17	−0.66	0.33
Conc Impulse [Ns]	158.16	32.33	160.99	32.62	**0.018**	0.58	0.04	1.10
Conc P Vel [m/s]	2.42	0.30	2.44	0.32	0.103	0.33	−0.18	0.83
Conc RFD/BM [N/s/kg]	21.56	15.32	20.84	17.01	0.458	−0.03	−0.52	0.46
Conc RFD [N/s]	1477.09	1004.82	1462.13	1319.38	0.487	−0.01	−0.50	0.48
Conc RPD/BM [W/s/kg]	205.20	62.70	216.72	77.21	0.224	0.20	−0.30	0.69
Conc RPD [W/s]	9223.37	4015.93	9223.37	5039.08	0.176	0.24	−0.26	0.73
F at P Power [N]	1325.75	167.75	1344.66	190.93	0.174	0.24	−0.26	0.74
RSI-modified [m/s]	0.32	0.10	0.36	0.13	**0.011**	0.64	0.09	1.17
Vel at P Power [m/s]	2.19	0.27	2.21	0.28	0.133	0.29	−0.22	0.79
Braking Phase Dur [s]	0.40	0.12	0.35	0.12	0.079	−0.37	−0.87	0.14
Ecc Braking RFD/BM [N/s/kg]	46.94	25.81	56.19	32.13	0.050	0.44	−0.08	0.95
Ecc Braking RFD [N/s]	3165.69	1524.04	3837.63	1964.34	0.057	0.42	−0.10	0.93
Ecc Dec Phase Dur [s]	0.21	0.07	0.19	0.07	0.113	−0.32	−0.81	0.19
Ecc Dec RFD/BM [N/s/kg]	67.97	37.59	76.19	43.57	0.170	0.25	−0.26	0.74
Ecc Dec RFD [N/s]	4615.25	2469.22	5204.31	2654.83	0.181	0.24	−0.27	0.73
Ecc M Braking F [N]	773.57	104.06	804.16	106.54	**0.045**	0.46	−0.07	0.96
Ecc M Dec F [N]	1048.35	184.63	1083.10	182.31	0.212	0.21	−0.29	0.70
Ecc MF [N]	679.09	88.73	682.41	87.73	0.110	0.32	−0.19	0.82
Ecc M Power/BM [W/kg]	4.91	0.82	4.91	0.99	0.494	0.00	−0.49	0.49
Ecc M Power [W]	340.72	80.21	339.31	71.94	0.479	−0.01	−0.50	0.48
Ecc PF/BM [N/kg]	20.94	3.28	21.53	3.57	0.252	0.17	−0.33	0.66
Ecc PF [N]	1448.81	309.19	1489.75	272.38	0.285	0.15	−0.35	0.64
F at Zero Vel [N]	1447.72	310.12	1486.53	270.77	0.294	0.14	−0.36	0.63

Bolded *p* values denote significant differences between conditions (*p* < 0.05). BM = Body mass; Conc MF = concentric mean force; Conc PF = concentric peak force; Conc M power = concentric mean power; FT = Flight time; CT = Contraction time; FT:CT = ratio between flight time and contraction time; JH = Jump height; P power = concentric peak power; Conc Dur = duration of the concentric phase; Conc P Vel = concentric peak velocity; Conc RFD = concentric rate of force development; Conc RPD = concentric rate of power development; F at P Power = Force at peak power; RSI-mod = Reactive strength index-modified; Vel at P Power = Velocity at peak power; Braking phase dur= braking phase duration; Ecc Braking RFD = rate of force development during braking phase; Ecc Dec Phase Dur = Duration of the eccentric deceleration phase; Ecc Dec RFD = rate of force development during deceleration phase; Ecc M Braking F = mean force during braking phase; Ecc M Dec F= mean force during deceleration phase; Ecc MF = Eccentric mean force; Ecc M Power = Eccentric mean power; Ecc PF = Eccentric peak force; F at Zero Vel = Force at zero velocity.

**Table 3 nutrients-13-01705-t003:** Kinetic and kinematic differences between Caffeine and Placebo conditions in the unilateral CMJ.

	Right						Left					
	Placebo		Caffeine				Placebo		Caffeine			
	Mean	SD	Mean	SD	*p*	ES	Mean	SD	Mean	SD	*p*	ES
Conc MF/BM [N/kg]	14.77	1.43	14.94	1.46	0.176	0.27	14.69	1.51	14.82	1.40	0.213	0.22
Conc MF [N]	1033.8	168.1	1045.3	165.1	0.181	0.26	1041.6	151.8	1053.4	164.6	0.159	0.28
Conc PF/BM [N/kg]	17.91	2.16	18.34	2.31	0.069	0.44	17.61	2.35	17.94	1.91	0.251	0.18
Conc PF [N]	1266.2	222.2	1289.7	232.5	0.087	0.40	1257.4	189.1	1284.9	211.5	0.221	0.21
Conc M Power/BM [W/kg]	13.66	3.82	14.34	3.14	0.125	0.34	13.54	2.92	13.93	3.16	0.199	0.23
Conc M Power [W]	972.2	335.3	1013.0	281.4	0.158	0.29	973.8	260.7	999.5	266.9	0.219	0.21
FT [ms]	341.50	53.82	349.00	61.24	**0.044**	0.52	335.79	55.52	340.93	54.56	0.182	0.25
CT [ms]	881.73	187.39	857.89	180.85	0.234	−0.21	937.39	174.78	880.46	185.12	0.078	−0.40
FT:CT	0.40	0.10	0.42	0.11	0.094	0.39	0.37	0.10	0.41	0.10	**0.040**	0.51
JH (Flight Time) [cm]	14.65	4.54	15.37	5.33	**0.039**	0.54	14.21	4.61	14.60	4.73	0.211	0.22
JH (Imp-Mom) [cm]	12.34	6.04	13.64	4.98	0.116	0.35	11.94	4.39	12.74	4.52	0.181	0.25
P Power/BM [W/kg]	25.37	6.95	26.76	5.86	0.131	0.33	24.63	4.71	25.65	5.21	0.098	0.36
P Power [W]	1809.1	624.9	1889.7	526.0	0.173	0.27	1773.3	453.2	1835.9	446.7	0.111	0.34
Conc Dur [ms]	320.81	54.36	327.39	60.93	0.310	0.14	326.89	51.69	325.71	60.86	0.470	−0.02
Conc Impulse [Ns]	107.94	34.76	113.66	29.90	0.098	0.38	108.51	28.67	111.09	26.45	0.243	0.19
Conc P Vel [m/s]	1.72	0.34	1.80	0.27	0.104	0.37	1.71	0.24	1.75	0.26	0.220	0.21
Conc RFD/BM [N/s/kg]	14.77	11.02	15.96	15.00	0.330	0.12	10.80	3.44	13.90	9.47	0.136	0.31
Conc RFD [N/s]	1017.6	662.3	1086.2	973.5	0.368	0.10	782.1	286.9	977.2	654.0	0.159	0.28
Conc RPD/BM [W/s/kg]	114.36	44.99	119.39	44.57	0.255	0.19	110.85	47.72	116.51	41.71	0.268	0.17
Conc RPD [W/s]	8055.0	3214.4	8346.8	3013.3	0.288	0.16	7821.8	3028.0	8302.0	2969.0	0.227	0.21
F at P Power [N]	1147.8	190.6	1157.8	182.6	0.239	0.20	1132.4	161.2	1156.3	167.0	0.082	0.39
RSI-modified [m/s]	0.17	0.07	0.19	0.07	**0.034**	0.56	0.16	0.06	0.18	0.07	**0.040**	0.51
Vel at P Power [m/s]	1.55	0.30	1.61	0.24	0.135	0.32	1.55	0.21	1.58	0.23	0.308	0.14
Braking Phase Dur [s]	0.37	0.11	0.36	0.11	0.396	−0.07	0.38	0.11	0.36	0.13	0.297	−0.15
Ecc Braking RFD/BM [N/s/kg]	36.85	25.57	33.73	20.48	0.217	−0.22	34.79	25.62	34.04	20.31	0.442	−0.04
Ecc Braking RFD [N/s]	2408.0	1255.4	2346.3	1299.9	0.400	−0.07	2419.7	1479.7	2430.7	1417.1	0.488	0.01
Ecc Dec Phase Dur [s]	0.21	0.06	0.20	0.06	0.061	−0.46	0.23	0.08	0.21	0.09	0.228	−0.21
Ecc Dec RFD/BM [N/s/kg]	36.08	23.14	37.69	25.08	0.329	0.13	37.36	32.62	38.68	26.63	0.421	0.05
Ecc Dec RFD [N/s]	2505.6	1529.3	2626.7	1612.9	0.314	0.14	2574.1	1954.3	2764.5	1821.3	0.340	0.11
Ecc M Braking F [N]	774.62	103.64	761.64	101.46	0.060	−0.47	795.02	103.05	782.24	116.14	0.159	−0.28
Ecc M Dec F [N]	955.07	142.82	948.30	170.99	0.346	−0.11	921.43	102.05	930.90	181.65	0.392	0.08
Ecc MF [N]	693.73	86.30	689.50	86.54	0.076	−0.42	704.86	94.05	703.46	94.71	0.324	−0.13
Ecc M Power/BM [W/kg]	3.70	0.51	3.54	0.85	0.203	−0.24	3.66	0.64	3.38	0.83	0.165	−0.27
Ecc M Power [W]	260.74	60.99	250.77	87.97	0.245	−0.20	259.25	52.79	243.64	78.50	0.204	−0.23
Ecc PF/BM [N/kg]	16.27	2.20	16.18	2.25	0.409	−0.07	16.46	2.76	16.23	2.36	0.352	−0.10
Ecc PF [N]	1149	214	1139	228	0.357	−0.10	1175.1	206.6	1167.9	251.7	0.432	−0.05
F at Zero Vel [N]	1145	213	1134	228	0.331	−0.12	1171.5	210.2	1164.4	250.5	0.432	−0.05

Bolded *p* values denote significant differences between conditions (*p* < 0.05). BM = Body mass; Conc MF = concentric mean force; Conc PF = concentric peak force; Conc M power = concentric mean power; FT = Flight time; CT = Contraction time; FT:CT = ratio between flight time and contraction time; JH = Jump height; P power = concentric peak power; Conc Dur = duration of the concentric phase; Conc P Vel = concentric peak velocity; Conc RFD = concentric rate of force development; Conc RPD = concentric rate of power development; F at P Power = Force at peak power; RSI-mod = Reactive strength index-modified; Vel at P Power = Velocity at peak power; Braking phase dur= braking phase duration; Ecc Braking RFD = rate of force development during braking phase; Ecc Dec Phase Dur = Duration of the eccentric deceleration phase; Ecc Dec RFD = rate of force development during deceleration phase; Ecc M Braking F = mean force during braking phase; Ecc M Dec F = mean force during deceleration phase; Ecc MF = Eccentric mean force; Ecc M Power = Eccentric mean power; Ecc PF = Eccentric peak force; F at Zero Vel = Force at zero velocity.

## Data Availability

Data presented in this study are openly available in FigShare. DOI: 10.6084/m9.figshare.14604531e.

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
