# Peer review of "Caffeine Doses of 3 mg/kg Increase Unilateral and Bilateral Vertical Jump Outcomes in Elite Traditional Jiu-Jitsu Athletes"

_nutrients, 2021, doi:10.3390/nu13051705_

Round 1

Reviewer 1 Report

This is an interesting study and adds to the existing literature supporting the ergogenic effects of caffeine on athletic performance. Since you had an even split of male and female athletes it might be wise to add sex as a covariate in your statistical analyses since sex does influence jump performance. Further comments are provided below.

Line 14: “athletes participated in two experimental trials”

Line 14: “double-blind”

Line 59: “Just one study has investigated the effects…”

Line 65: “…no research has established..”

Line 68: “3mg/kg of caffeine on the performance ….”

Why are you assessing the effects of 3mg/kg of caffeine and not 5 or 8? You need to provide a rationale. Also, please provide some rationales for your hypotheses. 

Line 108: “Each participant performed 3 CMJ…”

Lines 109-110: “Three minutes..”

Line 110: “Sixty minutes..”

Line 129: Full stop after “velocity”

Lines 120-136: You keep using both past and present tense. Please just use past tense.

Line 149: “Shapiro-Wilk”

Lines 156-157: The threshold for ES need to be checked. For example is an ES of 0.2 trivial or small?

Line 159: “both types of jumps”

Line 176: “Despite no significant…” – this appears incomplete?

Line 182: “..were higher..”

Please spell out RSI earlier in manuscript.

Line 199: “effects of 3mg/kg of caffeine”

Please add legends to all tables with meanings of all abbreviations.

Line 206: Make sure all abbreviations have been first spelled out prior to abbreviating earlier in the manuscript.

Line 213: “In those studies…”

Line 222: “shown ergogenic effects”

Line 237: “In parallel” – Please revise this word selection.

Line 254: “on the central neural system.”

Line 266: “Despite no statistically significant..” – This part of sentence is incomplete.

Line 276: “slight increase”

Line 300: “unilateral jump height”

Author Response

Response to Editor’s and Reviewer’ comments

Caffeine doses of 3mg/kg increase unilateral and bilateral vertical jump outcomes in elite traditional Jiu-Jitsu athletes

Dear Editor,

Dear Reviewer,

On behalf of my co-authors and myself, I would like to thank you for your valuable comments and for allowing us the possibility to re-submit our manuscript. Here, we have responded to the Reviewer’s comments and we have applied the necessary changes within the manuscript to improve the article. We believe that our manuscript has been improved by the suggested changes.

To ensure adequate English language the manuscript has been reviewed by an English editor. Invoice and proofreading certificate are enclosed. Changes made by the English editor are highlighted  in green along the manuscript.

Reviewer 1:

Reviewer comment: This is an interesting study and adds to the existing literature supporting the ergogenic effects of caffeine on athletic performance. Since you had an even split of male and female athletes it might be wise to add sex as a covariate in your statistical analyses since sex does influence jump performance. Further comments are provided below.

Authors’ response:  Dear reviewer, thank you for taking the time to review the manuscript and for your comments on it. The analysis by sex was not performed because splitting the sample would minimize statistical power and increase type I error below acceptable limits.

Line 14: “athletes participated in two experimental trials”

Authors’ response:  changed. Now it reads: “Jiu-Jitsu athletes took part in two experimental trials”

Line 14: “double-blind”

Authors’ response:  changed. Now it reads: “double-blind”

Line 59: “Just one study has investigated the effects…”

Authors’ response:  changed. Now it reads: “Just one study presents insights...”

Line 65: “…no research has established..”

Authors’ response:  changed. Now it reads: “no research has established the possible...”

Line 68: “3mg/kg of caffeine on the performance ….”

Why are you assessing the effects of 3mg/kg of caffeine and not 5 or 8? You need to provide a rationale. Also, please provide some rationales for your hypotheses. 

Authors’ response:  A rationale has been introduced for the use of 3mg/kg and for our hypothesis. Now it reads:Despite the observed performance enhancing effects of a low dose of caffeine on bilateral CMJ, the hypothetic potential of caffeine to improve single-leg jump performance and the absence of side effects in comparison to higher doses (i.e 6 and 9mg/kg)[16], to the authors´ knowledge, no research has established the possible enhancing effects of 3mg/kg of caffeine on bilateral and unilateral countermovement jump kinetics and kinematics in the different phases of the jump. Therefore, the aim of this research was twofold: i) to identify the effects of 3mg/kg on the performance and temporal outcomes of bilateral CMJ in all its phases and ii) to establish the possible ergogenic effects of this dose on performance and temporal metrics of the unilateral CMJ. We hypothesized that 3mg/kg of caffeine would increase the concentric metrics of both bilateral and unilateral CMJ kinetics and kinematics as thisdose appears to increase high velocity-low resistance actions[16].”

Line 108: “Each participant performed 3 CMJ…”

Authors’ response:  changed. Now it reads: “3 bilateral CMJ and 6 unilateral CMJ...”

Lines 109-110: “Three minutes..”

Authors’ response:  changed to number.

Line 110: “Sixty minutes..”

Authors’ response:  changed to number.

Line 129: Full stop after “velocity”

Authors’ response:  changed.

Lines 120-136: You keep using both past and present tense. Please just use past tense.

Authors’ response:  changed verbs to past tense.

Line 149: “Shapiro-Wilk”

Authors’ response:  changed.

Lines 156-157: The threshold for ES need to be checked. For example is an ES of 0.2 trivial or small?

Authors’ response:  changed. Now it reads: “Cohen´s d estimated magnitudes were calculated and classified in all between-groups comparisons [22].”

Line 159: “both types of jumps”

Authors’ response:  changed. Now it reads: “Most of the variables for bilateral and unilateral jumps...”

Line 176: “Despite no significant…” – this appears incomplete?

Authors’ response:  sentence completed . Now it reads: “Despite no significant results...”

Line 182: “..were higher..”

Authors’ response:  changed.

Please spell out RSI earlier in manuscript.

Authors’ response:  spelled in line 187.

Line 199: “effects of 3mg/kg of caffeine”

Authors’ response:  changed.

Please add legends to all tables with meanings of all abbreviations.

Authors’ response:  meanings of abbreviations included in the three tables.

Line 206: Make sure all abbreviations have been first spelled out prior to abbreviating earlier in the manuscript.

Authors’ response:  All abbreviations spelled previously,.

Line 213: “In those studies…”

Authors’ response:  changed. Now it reads: “In that research,...”

Line 222: “shown ergogenic effects”

Authors’ response:  changed. Now it reads: “doses ranging from 2-6 mg/kg of caffeine have shown ergogenic effects in the vertical jump...”

Line 237: “In parallel” – Please revise this word selection.

Authors’ response:  changed by “equally”

Line 254: “on the central neural system.”

Authors’ response:  changed

Line 266: “Despite no statistically significant..” – This part of sentence is incomplete.

Authors’ response:  changed. Now it reads: “Despite not being statistically significant...”

Line 276: “slight increase”

Authors’ response:  changed.

Line 300: “unilateral jump height”

Authors’ response:  changed.

Reviewer 2 Report

I believe the study design, methods, and approach to analysis of the study findings are all strengths.  

Despite the significant findings and in some cases, moderate effect sizes, the stated limitation by the authors of no mean differences exceeding the minimal detectable change reduces the practical value of these observations.  With this in mind, it seems the variability in the jumping of the participants (assuming because they Jiu-Jitsu athletes, who rarely jump) created some of the challenge for detecting meaningful change

However, since this is an ergogenic study, not a training study, I believe the authors statements are fair, considering the minimal dosage provided.  I would not have expected 

Overall, the manuscript needs to be reviewed and improved for grammar for publication in English (minor but consistent errors that are easily resolved).  

Additionally, It may be of value to "normalize" data against or at least speak to the differences in "sidedness" of the participants.  Would it be of value to look at the "preferred leg" unilateral changes for a better understanding of the effects/strategies of execution with and without caffeine.  I believe the authors have attempted this by using percent change

Thank you for doing a power analysis.. It strengthens the overall statements and ultimately provides a bit more confidence in the findings, despite the aforementioned concerns/points of suggestion

Author Response

Response to Editor’s and Reviewer’ comments

Caffeine doses of 3mg/kg increase unilateral and bilateral vertical jump outcomes in elite traditional Jiu-Jitsu athletes

Dear Editor,

Dear Reviewer,

On behalf of my co-authors and myself, I would like to thank you for your valuable comments and for allowing us the possibility to re-submit our manuscript. Here, we have responded to the Reviewer’s comments and we have applied the necessary changes within the manuscript to improve the article. We believe that our manuscript has been improved by the suggested changes.

Reviewer 2:

Reviewer comment: I believe the study design, methods, and approach to analysis of the study findings are all strengths.  

Despite the significant findings and in some cases, moderate effect sizes, the stated limitation by the authors of no mean differences exceeding the minimal detectable change reduces the practical value of these observations.  With this in mind, it seems the variability in the jumping of the participants (assuming because they Jiu-Jitsu athletes, who rarely jump) created some of the challenge for detecting meaningful change

However, since this is an ergogenic study, not a training study, I believe the authors statements are fair, considering the minimal dosage provided.  I would not have expected 

Overall, the manuscript needs to be reviewed and improved for grammar for publication in English (minor but consistent errors that are easily resolved).  

Additionally, It may be of value to "normalize" data against or at least speak to the differences in "sidedness" of the participants.  Would it be of value to look at the "preferred leg" unilateral changes for a better understanding of the effects/strategies of execution with and without caffeine.  I believe the authors have attempted this by using percent change

Thank you for doing a power analysis.. It strengthens the overall statements and ultimately provides a bit more confidence in the findings, despite the aforementioned concerns/points of suggestion

Authors’ response:  Thank you for your comments and your time to review this manuscript. To overcome the language issues a language editor proofread the manuscript. All the changes the  language editor made are highlighted in green. Invoice and proofreading certificate are enclosed. Regarding the preferred leg selection, it was decided to use the analysis by laterality (left/right) since the "dominant leg" is the leg that is placed behind in the stance and this position is variable during the fight.

Round 2

Reviewer 1 Report

Well done on addressing my comments and improving the manuscript.